# Testing Psychometric Properties and Measurement Invariance of Basic Psychological Needs in the Digital Version of the Sport Scale

Nuria Pérez-Romero [1], Rafael E. Reigal [1], María Auxiliadora Franquelo [1], Diogo Monteiro [2,3,4], Isabel Castillo [5], Antonio Hernández-Mendo [1] and Verónica Morales-Sánchez [1,*]

1   Department of Social Psychology, Social Work, Social Anthropology and East Asian Studies, University of Málaga, 29016 Málaga, Spain
2   ESECS, Polytechnic of Leiria, 2401-911 Leiria, Portugal
3   Research Center in Sport Sciences, Health Sciences and Human Development (CIDESD), 5000-801 Vila Real, Portugal
4   Life Quality Research Center (CIEQV), 2401-911 Leiria, Portugal
5   Department of Social Psychology, University of Valencia, 46010 Valencia, Spain
*   Correspondence: vomorales@uma.es

**Abstract:** Motivation is an important field in sport because it is related to the satisfaction, psychological well-being, or adherence to sport. The Psychological Need Satisfaction in Exercise Scale (PNSE) is one of the questionnaires that assess motivation from the Self-Determination Theory. Online tools are growing because of the advantages that they offer. The PNSE has been validated in different populations but never in its digital version. The aim of the present study was to analyze the psychometric properties of the digital version of the PNSE hosted on the MenPas platform. The current study included 1050 platform users aged 18 to 58 who engage in regular physical-sports activity. A confirmatory factor analysis (CFA) of the 18-item model was conducted, and invariance was performed according to gender and type of sport. The results indicated that the measurement model displayed a good fit to the data: (CFI = 0.93, TLI = 0.92, RMSEA = 0.08, SRMR = 0.06; $df$ = 132; B-S $p$ = 0.02–0.07): general sample ($\chi^2$ = 934.86, $\chi^2/df$ = 7.08), female ($\chi^2$ = 699.94, $\chi^2/df$ = 5.30), male ($\chi^2$ = 442.42, $\chi^2/df$ = 3.35) individual sports ($\chi^2$ = 753.17, $\chi^2/df$ = 5.71), and team sports ($\chi^2$ = 390.44, $\chi^2/df$ = 2.96). Appropriate values of invariance, convergent validity, discriminant validity, and composite reliability were obtained. The digital version of the PNSE shows adequate psychometric properties and it could improve the data collection process in future investigations.

**Keywords:** motivation; self-determination theory; sport; invariance; psychometric properties

## 1. Introduction

The human being has a natural inclination to move towards self-regulation, adaptation, and fulfillment in search of personal growth. Motivation is a concept that indicates for how long and where people direct their energy [1]. It is one of the most studied variables in the physical activity context, including sport and exercise [2]. There are several theoretical models that study sport motivation. One of them is the Theory of Self-Determination (SDT), which considers that people can direct their behavior in a level of controlled or autonomous way [3]. The SDT conceives people as active beings in constant growth, mastery of the environment, integration of experiences, and search for satisfaction of their basic needs [4,5]. It explains the factors that facilitate or hinder intrinsic motivation, autonomous motivation or well-being [6].

The collection of data in research is usually an arduous process for both professionals and participants. The recent adversities experienced due to COVID-19 affect daily lives and different habits such as physical activity and sport adherence [7], making it difficult for many people to gather in one place. Online tools could solve this and provide many advantages in different ways: they reduce the data collection time, offer immediate results,

facilitate questionnaire completion, and are cheaper and more sustainable [8,9], thus high-lighting the need for progress in digital data collection techniques [10–12]. Following this, Reigal et al. [13] found an increase in MenPas users during the pandemic. Therefore, it may be interesting to move the questionnaires to the digital format, to improve the storage and processing of data. For this purpose, there are already platforms that allow this type of evaluation, such as MenPas 1.0 [8,14], an online psychosocial assessment platform. However, the application of questionnaires in digital format could modify their psychometric properties, so care must be taken in the choice of instruments. In this sense, differences were found in the questionnaire reliability carried out online and in a written way due to the reduction of social desirability encouraged by anonymity [9,15]. Thus, González-Ruíz et al. [9] conducted a study with MenPas users and found that the results of anonymous users were more reliable than those of registered users. Reliability was related to the order and response time of the items. These variables are measured automatically by the MenPas platform and provide a solution to one of the drawbacks of online assessment platforms such as the control of external variables when performing assessments. This increases the importance of further research in the field of online assessment and the analysis of the psychometric properties of questionnaires in a digital form.

Thus, although the questionnaire has been validated in sport in the Spanish population, its online version has not been validated, and the invariance between sports (individual and team sports) has not been tested.

Therefore, the main objective of this study was to validate the online Psychological Need Satisfaction in Exercise Scale (PNSE), which was placed on the MenPas platform, with a large sample of different sport athletes. In that way, confirmatory factor analysis was performed using the 18-item model offered by the literature [16], and multigroup analysis was performed to explore its use in populations with different characteristics, so invariance analysis was performed according to gender and type of sport (individual/team). Finally, convergent, discriminant validity and composite reliability were analyzed. It was expected to find satisfactory results, improving the paper version reliability and actualize the results presented by Moreno-Murcia et al. [16]. This study could offer an online tool that improves the data collection process, reducing the time of execution, time of recollection, and economic costs in future research, and its properties improve its reliability.

## 2. The SDT and PNSE

The PNSE assesses motivation through self-determination theory. This theory suggests that human behavior can be intrinsically motivated, extrinsic motivated or unmotivated. Intrinsic motivation is the most autonomous motivation and allows the promotion of satisfaction with the simple execution of a task without the need of external stimuli only for one's own satisfaction [6]. This type of motivation can explain adherence to a regular habit or behavior and has been linked, among others, to health, psychological well-being, doing actions or practicing sport for fun, cohesion, and satisfaction in sport teams, as well as to a greater practice of physical activity [6,16–18]. Extrinsic motivation, on the other hand, requires some element that rewards the performance of a behavior and athletes in sport contexts or people in other contexts for their consequences [6]. This type of motivation has been associated with less well-being and greater discomfort [19]. Finally, non-motivation is a lower level of autonomy, determined by a lack of purpose [1,20,21]. In sport, athletes do not find any reason to continue practicing it, not knowing why they stay in it [6].

The SDT is a macro theory composed of six mini-theories: cognitive assessment theory, organic integration theory, causal orientation theory, goal content theory, basic psychological needs theory (BPNT), and relationship motivation theory [3]. Specifically, the BPNT defends the existence of basic psychological needs that are inherent, distinct, and universal and whose satisfaction is considered essential to achieve well-being, optimal growth, and development [22]. Although the theory is active and growing, it currently postulates that the three main basic psychological needs are autonomy, competence, and relatedness. The three needs must be satisfied and not frustrating to achieve the full

functioning of a person [22]. Autonomy refers to the ability to decide without external influence, feeling integrated with one's own actions, thoughts, and feelings. Competence refers to the perception of acting and obtaining the expected result, having opportunities to improve skills and knowledge. Relatedness alludes to the links you have with people close to you, the possibility of having support from those around you, and being relevant to them [3].

BPNT has important implications in different areas. In sport, competence and relatedness have been related to athlete identity and engagement [23,24]. Basic psychological needs have a strong impact in traditional sports athletes [25] and e-sport players [26], promoting self-development and wellbeing in both. A lack of motivation hinders both the well-being and the intention to continue playing [26]. Thriving is highly predicted by the satisfaction of basic psychological needs [27]. All of these variables are relevant in sport development, and its study can be used by coaches or psychologists to increase adherence, satisfaction, and wellbeing in sport practice.

The PNSE evaluates the satisfaction of basic psychological needs in the context of physical exercise or sport [16,28]. It was developed by Wilson et al. [28], who demonstrated that the questionnaire was reliable by obtaining Cronbach's alpha values between 0.70 and 0.90. Its fit indices were adequate in the Canadian population and female and male university students, of whom 74.7% exercised weekly. Later, Moreno-Murcia et al. [16] adapted and validated the PNSE to a sport context in the Spanish population, with an average age of 14 years, obtaining adequate factor loadings, an adequate model fit after correlating errors, and good internal consistency, except for the autonomy factor, which was below the recommended range, being 0.80 for competence, 0.73 for relatedness, and 0.69 for autonomy. Although it showed adequate results, it could be improved by increasing the sample size and the use of digital technologies in data collection. Then, the PNSE was placed on the MenPas 1.0 platform with the same items used by Moreno-Murcia et al. [16], but in an online version with five selective buttons for each item and an option to obtain immediate results. The questionnaire has a total of 18 items grouped into three dimensions: competence, autonomy, and relatedness (Appendix A). However, it has never been validated in its online version.

Another important aspect in terms of psychometric analysis is related to the equivalence in groups with different characteristics (e.g., sex, sport, age-groups). In terms of gender, Leyton-Román et al. [7] found that men obtained higher results in autonomy. Leyton et al. [29] studied gender differences in rural and urban samples from 18 to 64 years, and men had higher scores in competence and autonomy but results were not significant. Sabo et al. [30] examined cross-gender invariance in the Malaysian version of the PNSE, Wilson et al. [28] carried out invariance in Canadians, and Vlachopoulos [31] did the same in private fitness centers, and all of them obtained satisfactory results. Similarly, Gunnell et al. [32] studied invariance across two questionnaires: the PNSE and PNSE-PA (Physical Activity, changing the word "exercise" to "Physical Activity") and across a sample with and without osteoporosis. The results did not show a strong fit for both groups, indicating that more research is required to conduct studies comparing samples. In another study [33], it was observed that the type of sport used in physical education classes could affect the satisfaction of basic psychological needs. In this sense, it indicated that the need for autonomy or competence could be more satisfied by individual sports, while the need for relatedness would be more satisfied by group sports. However, no invariance was conducted.

## 3. Materials and Methods

### 3.1. Participants

A total of 1050 (694 female) Spanish MenPas 1.0 platform users participated in the present study. The ages varied from 18 to 58 years (M = 23.82, SD = 5.75); 82.6% completed higher education, 13% had middle studies, 7% average studies, 1.4% had primary education, and 2.3% had no education. All participants were regular physical-sports activity practition-

ers who usually practiced sport an average of 6 h per week (M = 6.10; SD = 4.46). The sports practiced by the participants varied, being individual (69.3%) or team (30.7%) sports.

### 3.2. Measure
Basic Needs Satisfaction

The Psychological Needs Satisfaction in Exercise Scale (PNSE) [28] in Spanish [16] was used in its online form. This scale consists in 18 items, which participants responded to based on a Likert scale from 1 ("false") to 6 ("true"). Posteriorly, the items were grouped into three factors (six item each), representing the three basic psychological needs: autonomy ("I feel that I can do exercises in my own way"); competence ("I am confident to do the most challenging exercises"); and relatedness ("I think I get along well with those I relate to when we exercise together") underlying self-determination theory [3]. Participants were asked to respond items according to their perception of "in my trainings . . . ".

### 3.3. Procedure

Data were collected through MenPas platform [8,14], between 1 September 2012 and 8 January 2021. Participants had to register on the platform by filling in sociodemographic data (name, sport practiced, gender, age, studies, profession, among others) and complete the PNSE scale. Access to the platform and the handling of the collected data (without personal information) could only be done by the person responsible for the application (one of the authors of this work). In addition, the ethical principles of the Declaration of Helsinki [34] were respected throughout the research process. The work was approved by the Ethics Committee of the University of Málaga.

### 3.4. Statistical Analysis

Descriptive statistics, including means, standard deviation and bivariate correlations, were calculated for all variables. Confirmatory factor analysis (CFA) through maximum likelihood (ml) in AMOS 23.0 was performed, considering the previous factor structure of the PNSE [16]. AMOS 23 (IBM, Armonk, NY, USA) and IBM SPSS Statistics, Version 23 (IBM, Armonk, NY, USA) were used for data analysis.

#### 3.4.1. Construct Validity

The measurement model adequacy was verified through the traditional incremental and absolute indexes: comparative adjustment index (CFI), Tucker–Lewis Index (TLI), root mean square approximation error (RMSEA) with a 90% confidence interval (CI), and the standardized mean square residual (SRMR) [35,36]. For the referred indexes, the following cut-off values were adopted: CFI and TLI $\geq$ 0.90 and RMSEA and SRMR $\leq$ 0.08 [35–37]. Furthermore, the internal consistency was calculated by estimating the composite reliability, considering 0.70 as the cut-off value; average variance extracted (AVE) to evaluate convergent validity, and we defined values >0.50 as the cutoff for acceptability. Discriminant validity was achieved when construct AVE values were larger than the squared correlations across constructs of the measurement model [36].

#### 3.4.2. Multigroup Analysis

The multigroup analysis was performed to check if the model showed a good fit among different groups. Therefore, multigroup analysis between gender and the type of sports was performed. According to Cheung and Rensvold [38] and Byrne [35], these criteria must be verified if the following conditions are observed: (1) the measurement model must be adjusted for each group; (2) four types of invariances must be examined: configural, metric, scalar, and residual. The invariance assumptions were verified through CFI differences (<0.01) as suggested by Cheung and Rensvold [38]. The invariance models were evaluated following the recommendations presented by Chen [39] as follows: (i) for metric invariance, a change in SRMR ($\Delta$SRMR) of less than 0.030 and a change in RMSEA ($\Delta$RMSEA) of less than 0.015 as support for model fit and (ii) for scalar invariance, a change

in SRMR (ΔSRMR) of less than 0.010 and a change in RMSEA (ΔRMSEA) of less than 0.015 as an indication of good invariance.

## 4. Results

### *4.1. Preliminary Analysis*

A preliminary analysis revealed no missing values and outliers in the general sample. The item-level descriptive statistics showed that no violation in terms of the normal distribution was observed since the values of skewness and kurtosis ranged from +2 to −2 and +7 to −7, respectively [35]. However, the Mardia's coefficient of multivariate kurtosis exceeded the expected values for the assumption of multivariate normality (>5). Therefore, a Bollen–Stine (2000 samples) bootstrap was employed for subsequent analysis [40].

### *4.2. Construct Validity*

Regarding internal consistency, Table 1 shows the values for composite reliability, as well as the factorial loads for each item. Composite reliability showed a good fit for all three dimensions, being 0.94 for competence, 0.89 for autonomy, and 0.81 for relatedness, that is, all dimensions had values greater than or equal to 0.70 [36].

**Table 1.** Factorial loads, error, and composite reliability of the PNSE.

| | General Sample | | Female Sample | | Male Sample | | Individual Sample | | Team Sample | |
|---|---|---|---|---|---|---|---|---|---|---|
| | λ | SE | λ | SE | λ | SE | λ | SE | λ | SE |
| Competence | 0.94 | | 0.95 | | 0.94 | | 0.95 | | 0.91 | |
| Item 1 | 0.75 * | 0.02 | 0.75 * | 0.03 | 0.76 * | 0.03 | 0.77 * | 0.03 | 0.63 * | 0.04 |
| Item 4 | 0.87 * | 0.02 | 0.88 * | 0.02 | 0.84 * | 0.03 | 0.88 * | 0.02 | 0.82 * | 0.03 |
| Item 7 | 0.85 * | 0.02 | 0.86 * | 0.02 | 0.82 * | 0.04 | 0.87 * | 0.02 | 0.76 * | 0.04 |
| Item 9 | 0.89 * | 0.01 | 0.88 * | 0.02 | 0.89 * | 0.02 | 0.88 * | 0.02 | 0.88 * | 0.02 |
| Item 11 | 0.90 * | 0.02 | 0.92 * | 0.02 | 0.86 * | 0.03 | 0.91 * | 0.02 | 0.85 * | 0.03 |
| Item 14 | 0.85 * | 0.02 | 0.87 * | 0.02 | 0.81 * | 0.03 | 0.87 * | 0.02 | 0.77 * | 0.04 |
| Autonomy | 0.89 | | 0.89 | | 0.88 | | 0.91 | | 0.85 | |
| Item 2 | 0.55 * | 0.05 | 0.51 * | 0.06 | 0.62 * | 0.07 | 0.59 * | 0.05 | 0.47 * | 0.11 |
| Item 5 | 0.81 * | 0.03 | 0.81 * | 0.04 | 0.79 * | 0.05 | 0.84 * | 0.03 | 0.72 * | 0.07 |
| Item 8 | 0.80 * | 0.04 | 0.81 * | 0.04 | 0.79 * | 0.07 | 0.82 * | 0.04 | 0.78 * | 0.09 |
| Item 12 | 0.80 * | 0.03 | 0.81 * | 0.04 | 0.76 * | 0.06 | 0.83 * | 0.03 | 0.72 * | 0.07 |
| Item 15 | 0.80 * | 0.03 | 0.81 * | 0.04 | 0.77 * | 0.07 | 0.82 * | 0.04 | 0.75 * | 0.08 |
| Item 17 | 0.77 * | 0.04 | 0.80 * | 0.05 | 0.74 * | 0.10 | 0.79 * | 0.05 | 0.74 * | 0.11 |
| Relatedness | 0.81 | | 0.80 | | 0.83 | | 0.83 | | 0.75 | |
| Item 3 | 0.37 * | 0.08 | 0.38 * | 0.10 | 0.36 * | 0.16 | 0.42 * | 0.09 | 0.26 * | 0.17 |
| Item 6 | 0.41 * | 0.07 | 0.35 * | 0.09 | 0.50 * | 0.13 | 0.45 * | 0.08 | 0.31 * | 0.16 |
| Item 10 | 0.68 * | 0.04 | 0.68 * | 0.05 | 0.68 * | 0.08 | 0.70 * | 0.05 | 0.63 * | 0.09 |
| Item 13 | 0.75 * | 0.03 | 0.75 * | 0.04 | 0.74 * | 0.06 | 0.77 * | 0.04 | 0.68 * | 0.07 |
| Item 16 | 0.82 * | 0.03 | 0.81 * | 0.03 | 0.86 * | 0.04 | 0.84 * | 0.03 | 0.80 * | 0.05 |
| Item 18 | 0.75 * | 0.03 | 0.73 * | 0.03 | 0.80 * | 0.05 | 0.78 * | 0.03 | 0.69 * | 0.05 |

Note. λ = standardized factorial loads; SE = standardized error; Composite reliability coefficient is in italics; * $p < 0.01$.

Table 2 shows the results for convergent and discriminant validity and internal consistency. A measure of convergent validity is indicated by the extracted mean variance (AVE) is ≥0.50 [36]. Thus, it was observed that the AVE values were optimal for competence (AVE = 0.74) and autonomy (AVE = 0.58) and close to the cut-off point for relatedness (AVE = 0.44). As a measure of discriminant validity, the square of the correlation between factors must be lower than the AVE value for each factor [36], showing optimal results for competence–autonomy (r = 0.57; $r^2$ = 0.34), competence–relatedness (r = 0.53; $r^2$ = 0.28), and autonomy–relatedness (r = 0.39; $r^2$ = 0.15).

Confirmatory factor analysis was performed to analyze the structure of the questionnaire based on the model already proposed in the literature [15]. This model showed six items for the variable Competence (1, 4, 7, 9, 11, and 14), six items for the variable

Autonomy (2, 5, 8, 12, 15, and 17), and six items for the variable Relatedness (3, 6, 10, 13, 16, 18), as shown in Figure 1. As can be seen in Table 3, the analysis showed an adequate adjustment for all indices since the values were between CFI and TLI $\geq$ 0.90 and SRMR and RMSEA $\leq$ 0.08 [35–37] except for TLI in the team sample, which was near the cutoff.

**Table 2.** Descriptive statistics and convergent and discriminant validity of the general sample.

| Variables | Mean | SD | AVE | 1 | 2 | 3 |
|---|---|---|---|---|---|---|
| 1. Competence | 4.75 | 1.01 | 0.73 | 1 | 0.34 | 0.28 |
| 2. Autonomy | 4.26 | 1.05 | 0.58 | 0.57 ** | 1 | 0.15 |
| 3. Relatedness | 4.37 | 0.87 | 0.43 | 0.53 ** | 0.39 ** | 1 |

Note. SD = standard deviation; AVE = average variance extracted. ** $p < 0.01$; below the diagonal, the correlations are shown and above, the square of the correlations ($r^2$).

**Table 3.** Model goodness of fit indices for the PNSE.

| Model | $\chi^2$ | df | $\chi^2$/df | B-S $p$ | SRMR | CFI | TLI | RMSEA | 90% CI |
|---|---|---|---|---|---|---|---|---|---|
| | | | Previous models | | | | | | |
| Wilson et al. (2006) [28] | 688.03 | 132 | - | <0.01 | 0.07 | 0.94 | - | 0.09 | 0.08–0.09 |
| Wilson et al. (2007) [41] | 334.36 | 132 | - | <0.01 | - | 0.92 | - | 0.10 | 0.08–0.11 |
| Moreno-Murcia et al. (2011) [16] | 222.62 | 129 | - | <0.01 | 0.08 | 0.91 | 0.91 | 0.04 | - |
| | | | Current study models | | | | | | |
| General | 934.86 | 132 | 7.08 | <0.001 | 0.06 | 0.93 | 0.92 | 0.08 | 0.07–0.08 |
| Female | 699.94 | 132 | 5.30 | <0.001 | 0.07 | 0.93 | 0.92 | 0.08 | 0.07–0.09 |
| Male | 442.42 | 132 | 3.35 | <0.001 | 0.07 | 0.92 | 0.91 | 0.08 | 0.07–0.09 |
| Individual | 753.17 | 132 | 5.71 | <0.001 | 0.07 | 0.93 | 0.92 | 0.08 | 0.08–0.09 |
| Team | 390.44 | 132 | 2.96 | <0.001 | 0.07 | 0.90 | 0.89 | 0.08 | 0.07–0.09 |

Note. $\chi^2$ = chi–square; df = degrees of freedom; $\chi^2$/df = chi–normalized square; B-S $p$ = level of significance Bollen–Stine Bootstrap (2000) samples; SRMR = Standardized Root Mean Square Residual; CFI = Comparative Fit Index; TLI = Tucker–Lewis Index; RMSEA = Root Mean Square Error of approximation; CI = confidence interval.

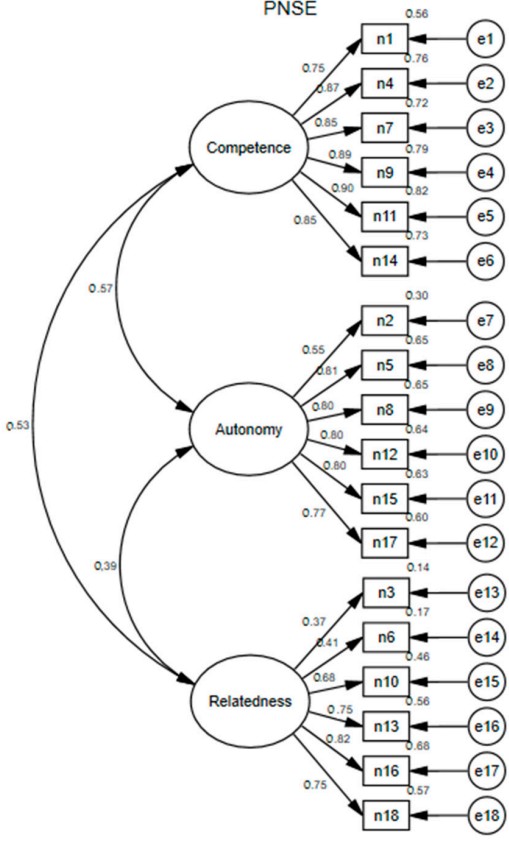

**Figure 1.** Standardized individual parameters of the PNSE for the general sample.

*4.3. Multigroup Analysis*

The analyzed model presented good adjustment indices for invariance both by gender and by type of sport, according to the indices shown by the authors [38,39]. As observed in Table 4, the results showed changes ≤0.01 in CFI (ΔCFI), ≤0.01 in RMSEA (ΔRMSEA), and ≤0.015 in SRMR (ΔSRMR) in all samples, so it can be assumed that the model is invariable in terms of gender and through the type of sport (team/individual), so this questionnaire can be used to make comparisons between samples through gender (male/female) and depending on whether the sport is an individual or team sport.

**Table 4.** Goodness of fit indices of invariance measures by gender and type of sport for the PNSE.

| Models | $\chi^2$ | *df* | $\Delta\chi^2$ | $\Delta df$ | *p* | CFI | ΔCFI | SRMR | ΔSRMR | RMSEA | ΔRMSEA |
|---|---|---|---|---|---|---|---|---|---|---|---|
| | | | | | Male–Female | | | | | | |
| CI | 1142.43 | - | 264 | - | <0.001 | 0.93 | - | 0.07 | - | 0.06 | - |
| MI | 1168.55 | 26.12 | 279 | 15 | <0.001 | 0.93 | 0.00 | 0.07 | 0.00 | 0.06 | 0.00 |
| SI | 1212.46 | 70.03 | 285 | 21 | <0.001 | 0.92 | 0.01 | 0.08 | 0.01 | 0.06 | 0.00 |
| RI | 1259.18 | 116.75 | 303 | 39 | <0.001 | 0.92 | 0.01 | 0.07 | 0.00 | 0.06 | 0.00 |
| | | | | | Individual–team | | | | | | |
| CI | 1143.68 | 264 | - | - | <0.001 | 0.93 | - | 0.07 | - | 0.06 | - |
| MI | 1172.62 | 279 | 28.94 | 15 | <0.001 | 0.93 | 0.00 | 0.07 | 0.00 | 0.06 | 0.00 |
| SI | 1203.13 | 285 | 59.45 | 21 | <0.001 | 0.92 | 0.01 | 0.07 | 0.00 | 0.06 | 0.00 |
| RI | 1347.80 | 303 | 204.12 | 39 | <0.001 | 0.91 | 0.02 | 0.07 | 0.00 | 0.06 | 0.00 |

Note. $\chi^2$ = chi–square; *df* = degrees of freedom; $\Delta\chi^2$ = differences in chi-square values; $\Delta df$ = differences in degrees of freedom; CFI = Comparative Fit Index; ΔCFI = differences in the Comparative Fit index s; SRMR = Standardized Root Mean Square Residual; ΔSRMR = differences in the values of the Standardized Root Mean Residual; RMSEA = Root Mean Square Error of Approximation; ΔRMSEA = differences in the values of the Mean Square Error of Approximation; CI = configural invariance; MI = metric invariance; SI = scalar invariance; RI = residual invariance.

**5. Discussion**

The main objective of this study was to analyze the psychometric properties of the PNSE in its digital version in a large sample of different athletes. For this purpose, confirmatory factor analysis was performed, and the internal consistency, convergent and discriminant validity, as well as invariance depending on gender and the type of sport (individual or team) were evaluated.

*5.1. Construct Validity*

The data offered in the model analysis provide an appropriate fit for the PNSE for all samples. First, the adjustment of the model for all samples presented favorable results with even better indices than previous studies based on non-digitized questionnaires [16,28,41]. Second, adequate values were found for convergent and discriminant validity, so the items were related to their respective factors coinciding with the structure of previous studies [16,28,41] and with the assumptions of the BPNT [3]. These results update those obtained previously and offer a better adjustment that allows the questionnaire to be used in its digital version. It resolves what was proposed by Moreno-Murcia et al. [16] on the possible modification of some items since the initial results were satisfactory with respect to the original questionnaire.

A good fit was also found in internal consistency, so all indicators evaluated the same construct [36]. Compared to the previous model, the results of consistency were better, increasing from 0.80 in competence, 0.69 in autonomy, and 0.73 in relatedness [16] to 0.94, 0.89, and 0.91, respectively, in the present study. These findings are congruent with the previous hypothesis based on the literature [8,14,15]. This has important implications for psychosocial assessment because it shows the improvement of psychometric properties in its tools, and future research may begin to apply this type of method for assessment.

*5.2. Multigroup Analysis*

A relevant aspect in the analysis of psychometric properties is the analysis of invariance [38,42] as it shows whether there is variability between groups. In the present study,

there was no variability between males and females or between individual sports and team sports. The invariance analyses were adequate, that is, the same items were associated with the same factors (configural invariance), the items were associated in the same way with each factor without changes in scores (metric invariance), the scores of both groups obtained the same unit of measurement and the same origin (scalar invariance) and, finally, the differences of the items of each group were only due to the factors (residual invariance). These results follow those of Wilson et al. [28], who analyzed invariance using sequential multigroup covariance analysis, obtaining partial invariance between genera. The study also reports new data from sport invariance, which was not done before. These results allow future research to compare samples of different types of sports.

### 5.3. Practice Applications

The results obtained offer favorable data and indicate that the PNSE can be used in its online version. As mentioned at the beginning, it is quite different from face-to-face assessments, and validation in the online context is increasing [11,43] because it brings many important benefits. It could help to collect more data in less time [8,14], enabling evaluators to obtain large samples for their studies and thus allowing for improved instrument validation. It also reduces the financial expense [10] so the money can be spent on other aspects of research and reduce the use of paper. Online evaluation speeds up the evaluation by obtaining results automatically and allowing an increase in scope; it can be done in different and distant places at the same time without the presence of the evaluator, reducing social desirability and human errors [8,14]. This study has benefited from all of these advantages, as the ease of data collection, and the breadth of the sample would never have been possible without using the MenPas platform. As the PNSE is used for the evaluation of motivation in teams or groups of individual athletes, coaches could use the MenPas platform from now on to assess it.

### 5.4. Limitations and Future Research

This research has some limitations. First, the online collection could generate some inaccuracy in the answers of some participants. However, the sample size reduces the possibility of bias in some responses. In future research, online tools need to be studied deeply, taking into account the variables mentioned at the beginning, such as the time or order of responses to the items or even creating new control variables such as pause buttons on the platform, indicating that the participant has had to stop due to a problem or distraction. Second, the analysis was not discriminated by age, which could lead to differences between population groups such as differences between adolescents and older adults. It is suggested that further studies should explore the data according to age and determine whether the psychometric properties are similar among different groups. Finally, it would be interesting to further analyze the different variables involved in the study, such as the influence of the education level or background, type of sport, age, gender or the impact of different time spans and their influence on motivation.

### 6. Conclusions

In conclusion, the data collected show that the questionnaire has a good psychometric fit in its digital version, even with higher reliability indices than previous paper studies. It satisfies the research objective and suggests that the online version of the questionnaire is a useful tool to assess the satisfaction of basic psychological needs in sport samples. Moreover, the settings show that the questionnaire can be taken by both males and females and among diverse types of sports. These results may become very important today due to the risk associated with the gathering of many people in the same space because of the changing COVID-19 situation and the need to advance in the development of digital techniques that contribute to scientific research [10,12]. This, in turn, could decrease the risk, time, and expenditure for both research and individual application by improving the efficiency of data collection [8,14]. This study yields results not analyzed before, offering more efficient,

more effective, and more sustainable alternatives in the motivation assessment in different types of sport.

**Author Contributions:** Conceptualization, I.C. and A.H.-M.; Data curation, M.A.F.; Formal analysis, N.P.-R., D.M. and A.H.-M.; Methodology, N.P.-R.; Project administration, R.E.R.; Resources, M.A.F. and V.M.-S.; Supervision, V.M.-S.; Validation, D.M.; Visualization, R.E.R.; Writing—original draft, N.P.-R.; Writing—review & editing, R.E.R., D.M., I.C. and A.H.-M. All authors have read and agreed to the published version of the manuscript.

**Funding:** This research received no external funding.

**Institutional Review Board Statement:** The study has been approved by the Ethics Committee of the University of Málaga. The indications of the Declaration of Helsinki [34], in the recommendations of a Good Clinical Practice of the EECC (document 111/3976/88 of July 1990), and the Spanish Legal Regulation in clinical research in humans (Royal Decree 561/1993) were followed.

**Informed Consent Statement:** Informed Consent (Law 15/1999 of December 13 on the Protection of Personal Data <LOPD>) appears in the conditions accepted when registering on the MenPas platform, so that users registered on the platform authorize the incorporation and processing of data as an electronic file owned by MenPas.

**Data Availability Statement:** Data are available on request from the authors.

**Conflicts of Interest:** The authors declare no conflict of interest.

## Appendix A

Escala de satisfacción de necesidades psicológicas en el ejercicio (PNSE).

Instrucciones: No hay respuestas correctas o incorrectas. Por favor, responde a las siguientes preguntas con sinceridad y Selecciona el número que representa mejor tu comportamiento, según la siguiente escala: 1. Falso. 2. Bastante falso. 3. Algo falso. 4. Algo verdadero. 5. Bastante verdadero. 6. Verdadero. En mis entrenamientos . . .

1.　Yo creo que puedo completar los ejercicios que son un reto personal.
2.　Siento que puedo hacer ejercicios a mi manera.
3.　Me siento atado a mis compañeros de ejercicios porque ellos me aceptan por quien soy.
4.　Tengo confianza para hacer los ejercicios más desafiantes.
5.　Creo que puedo tomar decisiones respecto a mi programa de ejercicios.
6.　Me siento como si tengo una obligación común con la genta que son importantes para mí cuando hacemos ejercicios juntos.
7.　Tengo confianza en mi habilidad personal de completar los ejercicios de mayor reto.
8.　Creo que yo estoy a cargo de las decisiones en mi programa de ejercicios.
9.　Creo que soy capaz de completar los ejercicios que me ofrecen el mayor reto personal.
10.　Siento una camaradería con mis compañeros porque hacemos ejercicios por la misma razón.
11.　Me siento capaz de completar los ejercicios más desafiantes.
12.　Creo que tengo voz en los ejercicios que hago.
13.　Me siento cercano a mis compañeros de ejercicios porque ellos saben lo difícil que pueden ser los ejercicios.
14.　Estoy contento en la manera en que puedo completar los ejercicios desafiantes.
15.　Creo que puedo escoger los ejercicios en que participo.
16.　Me siento relacionado con los que me relaciono cuando hacemos ejercicios juntos.
17.　Creo que soy el que decide los ejercicios que hago.
18.　Creo que me llevo bien con los que me relaciono cuando hacemos ejercicios juntos.

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
