# Peer review of "Testing Psychometric Properties and Measurement Invariance of Basic Psychological Needs in the Digital Version of the Sport Scale"

_sustainability, doi:10.3390/su141912126_

Round 1
Reviewer 1 Report
1. As soon as the samples of men's and women's, team and individual sports are compared, it is necessary to indicate in the theoretical part what differences are expected based on previous studies and theory, and in the results to show exactly what differences have turned out and discuss how much they meet expectations.
2. Stroke 210 – about normality
My remark: Normal distribution should be tested using Kolmogorov - Smirnov criterion. Also a normal distribution has the values of skewness and kurtosis lower than their 2*standard error of skewness and kurtosis correspondingly.
3. Stroke 242 – about models fitting
The models from the table 3 are not fit very well: CFI should be more then 0,95, RMSEA less than 0,05, χ2/df<2.
4. Stroke 249 (3.3. Multigroup) analysis and below
When doing multisampling comparison If there is a differences in the MI, the researcher should not test SI and RI. But having MI differences it is important to show for which items there is differences in factor loadings.
Reviewer 2 Report
|
Dear authors, I have been torn between recommending rejection and major revision. Unfortunately, after carefully reviewing the author's manuscript, I believe it currently does not meet the requirements for publication on Sustainability, even though the authors are relatively rigorous in the methodological section, their shortcomings are also explained in the "Limitations" section. The most difficult challenge to the publication of this manuscript, in my opinion, is the issue of innovation and scholarly merit. Another reason is that the authors appear to have failed to adequately demonstrate the necessity and significance of the PNES e-version. Finally, the writing style needs to be refined. The focus on the research question in the introduction and the expansion of the discussion section are likely important parts that can be improved. Setting aside the reasons for writing, the most challenging thing is probably articulating the academic contribution and innovation of this research.
Perhaps the following studies can serve as an inspiration and foundation for everyone. (1) doi.org/10.2196/16197 (2) doi.org/10.2196/24466 (3) doi.org/10.1371/journal.pone.0233891 (4) doi.org/10.3390/ijerph17134844 [this article has been cited]; The following information may be useful for the author to consider when revising the manuscript. I personally and sincerely suggest that the authors make major modifications to the introduction and discussion sections. Writing style requires extensive revision, particularly in terms of paragraph logic and sentence structure. The lack of logic and expression in the lines can easily cause distress to the reader and does not help the authors demonstrate their innovative ideas. The manuscript also appears to have many errors in the use of connecting words or to be illogical.
Abstract section Line 14. Personally, I believe that the authors should be encouraged to write the background from a professional standpoint, rather than relying on the current lack of digital version scales at PNSE. Line 15. It is recommended to add "the" before "present study". Line 16 "PNSE", "MenPas platform", and Line 25 "Psychometric Properties" are the same. Line 15. "Therefore," should be removed. Line 16-18. “A total of 1050 platform’s users, between 18-58 years who practice regular physical-sports activity participated in present study” might be considered to be rewritten as “The current study included 1050 platform users aged 18 to 58 who engage in regular physical-sports activity.” Line 18-20. “A Confirmatory Factor Analysis (CFA) of the 18-item model was performed, and gender and sport type invariance was tested.” In addition, Confirmatory Factor Analysis, the first letter appears to be capitalized unnecessarily. Line 24-25. Except for the issue of term capitalization, the conclusion is not very readable. Perhaps the authors should reconsider revising it.
Introduction section As a scholarly study, the manuscript is not clearly "problematic" (no offense intended). Perhaps the authors provided too much background information, and the most important aspect of this study (psychometric properties) lacks direct and focused description. This is very unfortunate.
Line 34-36. “There are a several theoretical models, however, in last decades the Theory of Self-Determination (SDT) has been used as motivation and personality theory who considers that people can lead their behavior in a level of controlled or autonomous way.” The terminology is unambiguous. However, it may be easy for the reader to attempt to flee.
Line 122-123. “The results did not show a strong fit for both groups and indicate that further research is needed to be able to perform studies comparing samples.” Perhaps it would be better to rewrite it as the following sentence? “The results did not show a strong fit for both groups, indicating that more research is required to conduct studies comparing samples.”
Line 130-132. I recommend merging into the previous paragraph. Could we reconsider rewriting it as“Thus, although the questionnaire has been validated in sport in the Spanish population, its online version has never been validated, and the invariance between sports (individual and team sports) has never been tested.” Line 133-134. “was the validation of the online PNSE, 133 placed on MenPas platform” to“was to validate the online PNSE, which was placed on the MenPas platform” Line 146- 151. 2.1 participants. 1) It is suggested that the recruitment process, inclusion criteria (sampling method), and demographic information (as stated in line 168 "sport practiced, gender, age, studies, profession") be reported in better detail. Are all participants of Spanish origin? Given that the authors claim that this study was conducted on "athletes," more information about the participants' characteristics may be useful. Because the PNES study will be of interest to exercise psychology and public health professionals, the authors should have defined this section more clearly. Undoubtedly, Wilson et al. (2006 IJSP) originally developed the PNSE based on students, which clearly illustrates the subject information (Study 1: 426 students; Study 2: 581 university students). 2)To what extent do different educational backgrounds influence the dependent variable?
Line 166. The time span is nearly a decade (from September 1, 2012 to January 8, 2021); may I inquire about the impact of such a long-time span on the results? Line 177. Maybe "studied" should be removed. Line 178. Please provide information about the AMOS 23.0 software. Although almost everyone is aware of it, we should all adhere to these basic guidelines.
Conclusion section Line 335. It's possible that the authors' claim that it's "even better than previous paper studies" is uncritical. Qualitative comparisons can be less or more rigorous. Line 341. If COVID-19 must be mentioned, I recommend that the authors include it in the introduction. This will help to provide critical support for the development and validation of digital scale.
Due to language barriers, I did not read Moreno-Murcia et al.’s study thoroughly. Herewith.
Best regards, Reviewer
|
Reviewer 3 Report
The article is logically structured and focuses on an interesting topic. There are, however, some issues that need to be improved.
Firstly, the writing needs to be improved (proof-reading preferably by an English native).
The introduction prepares the reader well, both from a theoretical point of view and regarding the need to carry out the study. However, the transition from SDT theories to online tools (lines 75 and 76) is abrupt. Especially with the word “otherwise”, which is incorrectly applied in the sentence. On the other hand, the existence of point 1.1 does not make sense to me, since the paragraph that starts this subsection follows precisely the previous paragraph. I suggest, therefore, to delete the title “1.1. present study”.
The introduction (all section 1) is a little too long. True, the information that is there is important and should be in the manuscript. However, I suggest that the authors to consider the possibility of making a shorter introduction (with a framing of the theme and formulation of the objectives of the study) and then a section 2 more dedicated to SDT and PNSE.
Regarding the methodological part, it is important to make the following observations: i) the PNSE with all the items should be available in annex; ii) what does “practiced some type of sport” mean? To better explain the type of sport and if the authors are not confusing it with physical activity, I suggest the authors to see, for example, the following article: “Contextual Factors Influencing Young Athletes’ Decision to Do Physical Activity and Choose a Sports’ Club” (2022).
The results appear consistent and the authors' objectives with the study are thus met. However, it is not clear why the analysis was not performed according to age.
I know what the authors state as their main objective (validating the scale), but what is the reason for not having analyzed the answers to the PNSE and only validating the scale?
Round 2
Reviewer 2 Report
Dear authors,
The revised manuscript is a significant improvement. All of my concerns were addressed, and I have no further suggestions.
In fact, I have already reviewed this manuscript and explained my decisions and revisions in detail. As a result, I do not believe I should continue to review this manuscript.
Reviewer 3 Report
The authors have replied to most of my remarks and made the necessary changes to improve the manuscript. However, the text really needs proof-reading to be publishable.